# Use of Cocktail of Bacteriophage for *Salmonella* Typhimurium Control in Chicken Meat

**DOI:** 10.3390/foods11081164

**Published:** 2022-04-17

**Authors:** Matías Aguilera, Sofía Martínez, Mario Tello, María José Gallardo, Verónica García

**Affiliations:** 1Departamento de Ciencia y Tecnología de los Alimentos, Facultad Tecnológica, Universidad de Santiago de Chile (USACH), Avenida Libertador Bernardo O’Higgins 3363, Estación Central, Santiago 9170022, Chile; matias.aguilera.b@usach.cl (M.A.); sofia.martinez@usach.cl (S.M.); 2Laboratorio de Metagenomica Bacteriana, Centro de Biotecnología Acuícola, Facultad de Química y Biología, Universidad de Santiago de Chile (USAC), Avenida Libertador Bernardo O’Higgins 3363, Estación Central, Santiago 9170022, Chile; mario.tello@usach.cl; 3Departamento de Medicina, Facultad de Medicina, Universidad de Atacama, Los Carrera 1579, Copiapó 1530002, Chile; mariajose.gallardo@uda.cl; 4Centro de Estudio en Ciencia y Tecnología de los Alimentos, Universidad de Santiago de Chile (CECTA-USACH), Obispo Manuel Umaña 050, Edificio de Alimentos, Estación Central, Santiago 9170201, Chile

**Keywords:** bacteriophages, food borne illness, poultry, *Salmonella*

## Abstract

Foodborne diseases are extremely relevant and constitute an area of alert for public health authorities due to the high impact and number of people affected each year. The food industry has implemented microbiological control plans that ensure the quality and safety of its products; however, due to the high prevalence of foodborne diseases, the industry requires new microbiological control systems. One of the main causative agents of diseases transmitted by poultry meat is the bacterium *Salmonella enterica*. Disinfectants, antibiotics, and vaccines are used to control this pathogen. However, they have not been efficient in the total elimination of these bacteria, with numerous outbreaks caused by this bacterium observed today, in addition to the increase in antibiotic-resistant bacteria. The search for new technologies to reduce microbial contamination in the poultry industry continues to be a necessity and the use of lytic bacteriophages is one of the new solutions. In this study, 20 bacteriophages were isolated for *Salmonella* spp. obtained from natural environments and cocktails composed of five of them were designed, where three belonged to the *Siphoviridae* family and two to the *Microviridae* family. This cocktail was tested on chicken meat infected with *Salmonella* Typhimurium at 10 °C, where it was found that this cocktail was capable of decreasing 1.4 logarithmic units at 48 h compared to the control.

## 1. Introduction

Foodborne illnesses continue to be the leading cause of deaths and hospitalizations worldwide, and among the pathogens associated with this type of illness, *Salmonella enterica* continues to be one of the most important ones, despite the microbiological control measures implemented by the industry [1,2,3,4,5]. It has been estimated that ≥26 million incident cases and 222,000 typhoid-related deaths occur annually [5]. *Salmonella* is mainly transmitted through the consumption of contaminated food, especially in poultry and poultry-derived foods [6,7,8]. To control these pathogens, poultry has used antibiotics during primary production. These compounds, together with reducing pathogens, produce an increase in the growth rate of birds when administered in sub-therapeutic concentrations, which caused this practice to quickly become widespread [9]. However, the increase in bacteria resistant to multiple antibiotics prompted health authorities to act and restrict the use of antibiotics during primary production [10]. As a consequence, the search for new antimicrobial treatments has been reactivated, where the use of bacteriophages is an increasingly attractive alternative, since they are based on natural phages without genetic modifications, do not contain preservatives, and are sold in aqueous solutions. On the other hand, they are low-cost and have been certified as GRAS, Kosher, and Halal [11,12].

Bacteriophages are viruses that attack bacteria, causing their lysis. They are abundant and found in all natural environments, with specific characteristics such as high specificity and the ability to self-replicate and limit themselves in the presence of their target bacteria. In addition to these characteristics, bacteriophages also have rapid antimicrobial action, making them a suitable alternative for use in the food industry [13]. This has been reported in food matrices such as chicken and fresh products and has been used to control different pathogens or spoilage microorganisms [14,15], as well as in different stages of production, pre-harvest, and post-harvest processes, among others. In this way, several marketing companies of products based on phage control for the food industry have been reported [16].

The aim of this study was to determine the effectiveness of a cocktail composed of five bacteriophages of *Salmonella* spp. in chicken meat at different temperatures to explore its use in this and other food matrices.

## 2. Materials and Methods

### 2.1. Bacterial Strains and Culture Medium

The bacterial host for the phages in this study was *Salmonella enterica* serotype Typhimurium ATCC14028. The *Salmonella* strains used for the host range are listed in Table 1. The strains were stored at −80 °C in Luria-Bertani broth (LB) supplemented with 50% *w*/*v* of glycerol, and a working stock of every strain was stored at 4 °C for up to 1 month. For each experiment, the bacteria were grown in 5 mL of Luria-Bertani broth supplemented with salt (supplemented with 1 mM CaCl_2_, 10 mM MgSO_4_) for 16 h at 37 °C with 100 rpm agitation. All experiments involving the manipulation of *Salmonella enterica* serotype Typhimurium ATCC14028 were performed in the CECTA laboratory under the ethical and biosafety standards of the Institutional Ethics Committee of the Universidad de Santiago de Chile (335) and current legislation on the subject (1613328729).

### 2.2. Bacteriophage Isolation

The bacteriophages used in this study were isolated from two sources, irrigation water (n = 1) and poultry feces (n = 36) both from the metropolitan region of Santiago, Chile. Prior to their isolation, the phages were enriched by mixing 10 g of samples, 90 mL buffer SM (sodium chloride, magnesium sulphate buffer: Tris-HCl 50 mM, NaCl 100 mM, MgSO_4_·7 H_2_O 8 mM pH 7.5), and 1 mL of S. Typhimurium ATCC14028 cells grown to 16 h, followed by incubation at 37 °C for 22 ± 2 h. The phage-enriched cultures were then spun at 4000× *g* and the supernatant was subsequently sterilized using a 0.45 μm filter membrane. Then, 100 μL of the filtrate was mixed with 300 μL of 1:10 diluted S. Enteritidis previously grown for 16 h and 4 mL of top agar (Luria-Bertani, agar-agar 0.7% *w*/*v*), according to the double agar overlay method [17,18]. Plates were incubated at 37 °C for 18 ± 2 h or until lyses plaque were visible. Single plaque isolation was carried out to obtain a pure phage lysate by lifting each one from the agar surface using a pipette tip, and resuspending the plaque in 200 μL SM buffer (Tris-HCl 50 mM, NaCl 100 mM, MgSO_4_·7 H_2_O 8 mM pH 7.5). Finally, phages were concentrated and stored at 4 °C for further analyses.

### 2.3. Host Range

The phage host-range was tested by spotting 5 µL of the lysate (10^5^ PFU/mL), in duplicate, on a lawn of bacteria host cells grown to 16 h at 37 °C. Plates were incubated at 37 °C for 16 h. The clearance zones were characterized with a scaling system as described by Kutter (2009), where no color indicated a zone with complete turbidity (no lysis) and orange indicated a completely clear zone with no turbidity. These experiments were performed in the laboratory of Dra. Andrea Moreno-Switt at Universidad Andres Bello, Chile. The serovars used were: Anatum FSL A4-525, Weltevreden FSL R8-798, Virchow FSL S5-961, Saintpaul FSL S5-369, Choleraesius FSL R9-1343, Corvalis FSL R8-092, Mbandaka FSL A4-793, Javiana FSL S5-406,Stanley FSL S5-464, Braenderup FSL S5-373, Infantis FSL S5-506, Typhimurium FSL A4-737, Montevideo FSL S5-474, Muenster FSL S5-917, Agora FSL S5-667, Kentucky FSL S5-431, Heidelberg FSL S5-455, Dublin FSL S5-368, Cerro FSL R8-370, Newport FSL S5-515, 4,5,12, i:-FSL S5-390, Enteritidis SARB16, Enteritidis SARB17, Enteritidis SARB18 Enteritidis SARB19, and Typhimurium ATCC 14028.

### 2.4. Lysis Kinetics

Culture S. Typhimurium ATCC14028 strains were incubated under the above conditions. The culture was then loaded into 96-well plates at a concentration of 1 × 10^5^ CFU/mL and infected with phage at MOI of 1, 0.1, or 0.01. Plates were placed into a plate reader at 37 °C and the cellular density was evaluated at OD_620nm_ every 20 min. This procedure was repeated three times.

### 2.5. One Step Growth

In order to determine the growth kinetics of phages, one-step growth experiments were performed following the procedure described before [19].

### 2.6. Electron Microscopy

The structure of the phage lysates with higher titer (10^9^–10^11^ PFU/mL) was analyzed by transmission electron microscopy (TEM), according to the procedure described by Deveau et al. [20]. Briefly, 1 mL of lysate was centrifugated at 4 °C for 1 h at 25,000× *g*. The supernatant, 900 μL, was gently removed and the pellet containing the phages was resuspended in the remaining 100 μL and mixed with 1 mL of 0.1 M ammonium acetate. Then, phages were collected by centrifugation again at 4 °C for 1 h at 25,000× *g*. This procedure was repeated twice before the last 100 μL were used for TEM. The carbon coated formvar and carbon grid was prepared using 15 μL of purified lysates. Then, the phage preparations were negatively stained with 15 µL of uranyl acetate (1% *w*/*v*) for 1 min, dried for 5 min at 55 °C, and analyzed using a Thermo Scientific™ Talos™ F200C, Waltham, MA, USA transmission electron microscope (UMA, Pontificia Universidad Católica de Chile Microscopy services). The image analysis was performed using the software Image J, Bethesda, MD, USA.

### 2.7. DNA Extraction and Restriction of Enzyme Digestion

The DNA samples were prepared as was described by Pickard (2009) [21]. The nucleic acid of each phage was digested with DNase I, RNase A, and Nuclease S1, and the genetic materials of phages with dsDNA were digested with EcoRI, HindIII, and HinfIII at 37 °C. Products of digested nucleic acid were separated by 1.5% *w*/*v* agarose gel electrophoresis.

### 2.8. Assay in Chicken Meat

Salmonella ATCC14028 inoculum was prepared as mentioned above. Packed chicken breasts were acquired in a local supermarket and were aseptically cut into pieces in the laboratory. To reduce the indigenous bacteria level before inoculation, the pieces of chicken breasts were irradiated with UV light for 30 min in a safety cabinet (15 min each side). Each meat sample was inoculated with 1 mL inoculum (∼4 log CFU/g of *Salmonella* ATCC14028) on the meat sample, 500 µL for each side, then left for 30 min at room temperature (for bacterial attachment to the samples). Inoculated meat samples were immersed for 5 min in 100 mL of SM buffer containing 10^9^ PFU/mL of each phage. Control samples were immersed in sterile SM buffers. As a negative control, non-inoculated samples were included to evaluate the background presence of *Salmonella*. The samples were stored at 10 °C, 22 °C, or 30 °C, and microbiological analysis was carried out at 24, 48, and 72 h. Each sample was homogenized with 90 mL of sterile 0.1% *w*/*v* peptone water in a stomacher at 100 rpm for 1 min. Then, 1 mL of homogenized samples was centrifuged at 10,000× *g* for 3 min at 4 °C, and the supernatant-containing phages was collected in another tube with several drops of chloroform. The pellets were re-suspended in 1 mL of sterile 0.1% *w*/*v* peptone water. The target bacteria were quantified by spread plates with serial dilution on selective media XLD agar plates and incubated at 37 °C for 48 h. *Salmonella* counts were converted to log CFU/g. Duplicate samples were included for each treatment and each storage period.

### 2.9. Statistical Analysis

All results were analyzed using GraphPad Prism 7.0 software. For normally distributed samples, an analysis of variance (ANOVA (α = 0.05)) and Bonferroni’s comparison tests were applied.

## 3. Results

### 3.1. Bacteriophage Isolation

Forty-one samples were used for the Salmonella phage isolation. From the total samples, 20 phages were obtained by enriching with S. Typhimurium ATCC14028; 19 of the isolated phages came from various poultry feces including turkey, chicken, and goose, and the remaining isolated phage was obtained from a water sample. These phages were used for general characterization and five of them were selected for the design of a cocktail to be used in chicken meat.

### 3.2. Characterization of Phages

#### 3.2.1. Host Range and Selection of Phage

The host range analysis of the phages showed that they were capable of lysing between 1 and 10 of the 26 strains of the *Salmonella* tested (Table 1). All the selected phages could infect the S. Typhimurium ATCC14028 strain, as it was the strain that was used to enrich the samples. Despite this, some phages did not infect the S. Typhimurium strain of the ILSI collection analyzed in the host range (FSL A4-737). In addition to the aforementioned strains, five strains of S. enteritidis were analyzed, four of them belonging to the SARB collection. We observed that 4 of the 20 phages infected the SARB17 and SARB19 strains and that the SARB18 strain was not infected by any of the analyzed phages.

Nine of the phages were classified as “narrow” host-range phages as they infected only the tested *Salmonella* Typhimurium strain ATCC14028 or some strains of *Salmonella* Enteritidis that were analyzed. It was noted that the phage A10 could infect most strains of *Salmonella* (corresponding to 10 of the 26 strains).

On the other hand, 53.8% (14/26) of the strains were resistant to all the phages analyzed (Anatum FSL A4-525, Weltevreden FSL R8-798, Virchow FSL S5-961, Saintpaul FSL S5-369, Corvalis FSL R8-092, Mbandaka FSL A4-793, Stanley FSL S5-464, Braenderup FSL S5-373, Montevideo FSL S5-474, Muenster FSL S5-917, Agora FSL S5-667, Kentucky FSL S5-431, Heidelberg FSL S5-455, Dublin FSL S5-368, and Enteritidis SARB18) The strains that were lysed by the highest number of phages were S. Typhimurium ATCC14028 (20/20), S. Coleraesius and Infantis (10/20), and *S*. Cerro and Typhimurium ILSI (9/20) (Table 1). From the total phages obtained and evaluated in their host range, five were selected for their characterization; the criterion used for the selection was their antimicrobial activity against S. Typhimurium and their high production of bacteriophages.

#### 3.2.2. Microscopy

Structural analysis by transmission electron microscopy showed that these phages differed in morphology. Phages A7, A8, and B3 belonged to the *Siphoviridae* family (Figure 1), all of which had an icosahedral-shaped head, non-contractile tails, and double stranded DNA. A7 was 83.1 ± 3.8 nm in head diameter with a 192.7 ± 6.6 nm long tail. A8 was 82.3 ± 1.7 nm in head diameter and 188.5 ± 9.7 nm in tail length, and phage B3 was 81.4 ± 2.3 nm head diameter and 191.0 ± 10.8 nm in tail length. On the other hand, phages A5 and A4 were spherical in shape and did not have a tail, which is consistent with the classification as *Microviridae*, with a size of 33.0 ± 4.0 nm and 59.6 ± 2.6 nm in diameter, respectively.

The genetic material of phages A7, A8, and B3 corresponded to a dsDNA, which was expected for Siphoviridae phages (Figure 2). On the other hand, phages A4 and A5 had an ssDNA genome, which was verified by the absence of digestion with DNAse I and RNAse A and digestion with nuclease S1 (data not shown). This is consistent with microscopy images showing small spherical capsids (Figure 1) and no tail.

On the other hand, the digestion profile with the enzymes EcoRI, HindIII, and HinfI showed differences in the phage profile, although with this method, it was not possible to differentiate phages A7 and B3. However, the morphology and microbiology data indicated that they were different bacteriophages.

### 3.3. One-Step Growth and Lysis Kinetics

The infection potential of each phage was analyzed by single-step growth curves (Figure 3, Table 2). A4 and A5 showed the shortest latency periods of approximately 20 min, while A8 showed the greatest latency of 100 min. The burst size of phages per infected cell varied between 0.8 (phage B3) and 1.8 (phage A4) phages per infected cell.

The kinetics of lysis indicated that the phages that released more viral progeny had a lower impact on the density of bacteria A4 and A5 (data not shown). On the other hand, phages A7 and B3 had the smallest burst size but a strong impact on the growth kinetics of bacteria (data not shown).

### 3.4. Salmonella Reduction in Chicken Meat

In our study, a cocktail of five phages achieved a 1.7 log10 CFU/g reduction in S. Typhimurium in 48 and 72 h of storage at 22 °C (Figure 4), indicating a strong bactericidal effect of our phage cocktail against S. Typhimurium on chicken meat. On the other hand, the phage cocktail achieved a 1 log10 CFU/g reduction in S. Typhimurium in 24 h storage at 22 °C and 30 °C. This assay could provide information on the behavior of this cocktail in the presence of a food matrix that was marketed at other temperatures, since the food matrix affects both the growth of the bacteria and the interaction between the bacteria and the bacteriophage. These experiments were stopped before the test at 10 °C due to the high state of decomposition of the meat.

The efficiency of the phage cocktail treatment to control *Salmonella* contamination in chicken meat was evaluated using an in-depth assay. Immersion of *Salmonella*-inoculated chicken breast fillets in bacteriophage solution significantly (*p* < 0.0001) reduced *Salmonella* by 1.5 and 1.2 log10 CFU/g on 48 and 72 h of storage at 10 °C, respectively, as compared to the untreated positive control (Figure 5). No *Salmonella* was detected from the negative control samples.

## 4. Discussion

The need for new methods of microbiological control applicable to food matrices is a current and required need for the elimination of pathogens, control of bacteria resistant to antibiotics, and increasing the shelf life of foods. In this sense, lithic phages are an alternative that can be applied directly to food or during food production as disinfectants, due to their stability under abiotic conditions, null toxicity, and selectivity in antimicrobial activity. The safety of bacteriophage-based bio-controllers is reflected in the fact that several of these preparations, which do not have preservatives or additives, have been Kosher- and Halal- certified and approved for use in organic food [22].

In this study, 20 Salmonella phages were isolated, of which five were selected for their high viral progeny and rapid effect in *Salmonella* Typhimurium cultures. In addition, phage A7 and A8 had antimicrobial activity in the Cerro FSL R8-370 serovar and phage A7 had antimicrobial activity against the S. Enteritidis strain SAB 17. The ILSI and SARB collections were chosen to evaluate the host range because they cover the main Salmonella serovars that affect the food and veterinary industry (specifically poultry). The host range results showed that the bacteriophages were found to only affect a small number of strains, noting that some different strains of the same serovar were not infected by the same phages; therefore, they were highly specific in their action. The five selected bacteriophages only infected S. Typhimurium and one of them (A7) infected S. Enteritidis, which together are the main cause of food-related outbreaks, so they were attractive candidates to be applied in food matrices [23].

The samples used to obtain the bacteriophages were wastewater and feces from different poultry, due to the fact that the bacteriophages are close to their hosts and Salmonella spp. is an inhabitant of the digestive tract of all animals and through it deposits phages in the feces and contaminated waters [24].

Among the bacteriophages chosen to make this cocktail of phages against *Salmonella*, three corresponded to the *Siphoviridae* family (A7, A8, and B3), observed by microscopy the icosahedral head and a long tail. The other two battery phages (A4 and A5) had a head without a visible tail and their genetic material was ssDNA. According to the International Committee on Taxonomy of Viruses (ICTV), these phages were from the *Microviridae* family, which only represents 2% of viruses, since 88% of all recognized phages belong to the order *Caudovirales*, which includes the family Siphoviridae [25,26].

The results obtained in the application of a bacteriophage cocktail at 10 °C showed a decrease of up to 1.5 log10 CFU/g at 24 h in the *Salmonella* Typhimurium count. Similar results have been obtained by other authors, although the temperature of the analysis varied between 4 and 10 °C. For example, Fiorentin et al. [27] reported a decrease of less than 1 log10 CFU/g unit of *Salmonella* Enteritidis after 3 days of storage, reaching more than one logarithmic unit after 9 days of storage at 5 °C [27].

On the other hand, Hungaro et al. [28] reported a reduction of 1 log CFU/cm^2^ unit of Salmonella Enteritidis in chicken skin when treating it by immersion in a suspension of 10^9^ PFU/mL bacteriophage for 30 min. Sukumaran et al. [29]. reported a decrease of 0.7 to 0.9 log10 CFU/g of *Salmonella* when immersing chicken breast fillets in a suspension of SalmoFreshTM (a commercial product containing six different phages) at a concentration of 109 PFU/mL for 20 s. Hungaro et al. [28] showed that the *Salmonella* count decreased by 1 log 10 CFU/cm^2^ unit after 30 min of immersion in a suspension in a cocktail of five phages [28]. On the other hand, Higgins et al. [30] showed a between 40% and 60% reduction of positive samples for *Salmonella* after a 2 h treatment with a phage cocktail and contamination with low doses of *Salmonella* (between 30 and 20 CFU) [30]. Bacteria counts at short post-treatment periods are an alternative treatment for chicken carcasses that can be used before freezing. Together, these investigations show that bacteriophage solutions can be used in the meat industry, enabling the reduction of Salmonella under conditions of production and distribution.

An important part of the chicken meat production is distribution and refrigeration. In this process, it is important to maintain a low temperature during the slaughter, processing, and transport of the meat, which is not always fulfilled mainly during the distribution of the meat to retail and storage in retail [31]. These fluctuations in temperature can allow the growth of *Salmonella* and the consequent health problems in the population [32]. Therefore, it is important to design a treatment for this type of meat sale and use antimicrobials that work in a wide temperature range.

In this study, the bacteriophage cocktail in chicken meat was evaluated at 22 and 30 °C; under these conditions, the decrease in bacterial load reached 1.5, 1.7, and 1.8 log10 CFU/g at 24, 48, and 72 h at 22 °C, respectively, and a log10 CFU/g at 30 °C at 24 h. It is important to note that chicken meat is not marketed at these temperatures [33], but these data are a useful approximation of the behavior of the bacteriophage cocktail in food matrices that are marketed at these temperatures. The low antimicrobial activity of the phage cocktail observed in these conditions could be explained by the high growth rate of *Salmonella* at 22 and 30 °C, for which we could analyze shorter incubation times as previously reported [15,25,34]. Previous studies also have shown significant reductions of salmonella in chicken breast treated with bacteriophages and stored at 25 °C. However, the evaluation times were of a maximum of 5 h, during which the bacterium increased only 1 logarithmic unit. Thus, Huang et al. [15] reported that the decrease in *Salmonella* Typhimurium due to treatment with the bacteriophage LPST10 reached between 1.9 and 4 logarithmic units after incubations for 6 h at 28 °C and MOI of 100. These results varied depending on the food matrix [15]. Similar results were obtained by Li et al. (2020) who used phage D1-2 in liquid egg and was able to verify its antimicrobial activity at 25 °C, observing the decrease of approximately 1 log unit after 24 h. Sritha and Bhat [34] described similar results in different food matrices and different temperatures, and observed at 28 °C a decrease of up to 4 log units of *Salmonella* spp. using a four-phage cocktail. These results depended on the food matrix and were evaluated at times under 24 h. In summary, the results obtained in this study are consistent with other studies, due to the fact that a decrease in the bacterial load of *Salmonella* spp. was observed in food matrices at temperatures around 20–30 °C.

The effect of temperature on the lytic activity of bacteriophages has previously been described in bacteriophages from different bacteria. An example of the above was described for Escherichia coli phages applied to milk, observing that the ECPS-6 phage had greater lytic activity at 25 °C than at 4 °C, which was not reflected in a difference in the rate of growth of the bacteria over short periods of time. However, at longer times, it was observed that at 25 °C, the bacterium had a greater growth at 25 °C than at 4 °C, showing a greater lytic activity of the ECPS-6 phage at 25 °C. These results suggest that bacteriophages showed higher lytic activity when the bacterium was metabolically active, which may be explained by the requirement of bacteriophages for the metabolic machinery of the bacterium for the production of viral progeny [35]. On the other hand, the results of our experiments showed that the lytic activity of the phage cocktail at 22 °C eliminated 1.7 logarithmic units of Salmonella. This result differs from that which was observed at 30 °C (it only managed to reduce 1 logarithmic unit), despite the fact that a greater increase in the bacterial population was observed at 30 °C compared to 22 °C in the first 24 h. This effect of temperature on the lytic activity of bacteriophages has previously been reported. Thus, Hammerl et al. (2021) described that *Yersinia enterolitica* and *Yersinia pseudotuberculosis* bacteriophages can lyse at 37 °C but not at 28 °C, the optimal growth temperature suggested for these bacterium [36]. A possible explanation for this behavior is that the bacteriophage receptors were available in greater quantity at 37 °C and not at the optimum temperature for bacterial growth. Another example was described with *Listeria monocytogenes* phages A511, LP-125, and LP-048. In this case, the bacteriophages did not show lytic activity at 37 °C, unlike what was observed at 30 °C. In this case, it should be considered that *L. monocytogenes* can grow in a broad range of temperatures (7–37 °C). However, this does not necessarily express the receptors required for phage action in this entire range. Furthermore, the authors showed that this effect could not be explained solely by a difference in adsorption efficiencies, suggesting that it was not only due to differential receptor availability at different temperatures [37]. Taken together, these observations of the effect of temperature with the lytic action of the phage are important for the design of biocontrollers applied to different food matrices in which Salmonella is a problem, because the conditions of the distribution and marketing chain of each food must be considered to evaluate the functioning of the bacteriophages under these conditions. This is highly relevant because the ubiquity of this microorganism has made multiple foods, both of animal and vegetable origin, vehicles for the transmission of this bacterium [38]. Thus, an antimicrobial solution capable of controlling this microorganism in the presence of food matrices and at different temperatures (such as 22 °C) is an opportunity for the production chain of other foods such as vegetables, fruits, eggs, and their derivatives. This has been proposed by other authors, who have evaluated the lytic activity of bacteriophages in matrices such as milk, egg yolk, and RTE lettuce, among others [15,25,39]. Furthermore, the industry requires disinfectant solutions during food production for the disinfection of countertops and utensils in order to avoid cross-contamination due to the formation of biofilms [40] or during the rearing of the animals to avoid contaminating the slaughter stage. The requirements to apply a cocktail of bacteriophages in the different stages of food production require that they be efficient in the elimination of *Salmonella* spp. at different temperatures, in addition to maintaining its action in different matrices or niches (such as the intestine of animals [41]) for which new experiments are required to determine other applications of the cocktail described in this work.

## 5. Conclusions

In conclusion, our cocktail of five of the phages isolated in this study allowed the decrease of S. Typhimurium in chicken meat at different temperatures. However, more work needs to be done to determine their suitability for use in the food industry as biocontrol agents and therapy agents.

## Figures and Tables

**Figure 1 foods-11-01164-f001:**
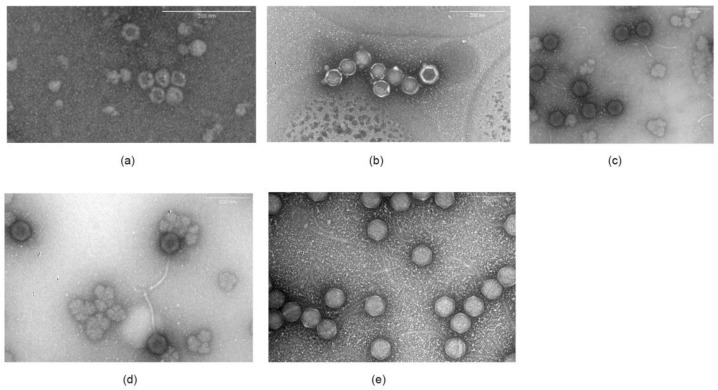
Electronic microscopy of transmission of phages (**a**) phage A.4 (**b**) phage A5, (**c**) phage, A7, (**d**) phage A8, and (**e**) phage B.3.

**Figure 2 foods-11-01164-f002:**
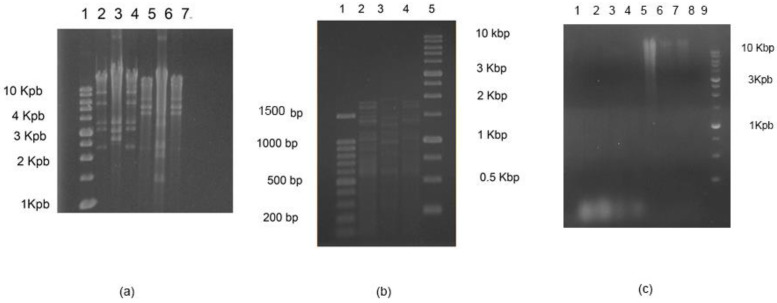
Genetic material of phages. (**a**) Profile of digestion with EcoRI and HindIII. Line 1: standard of molecular weight 1 kpb; line 2: DNA of phage A7 with EcoRI; line 3: DNA of phage A8 with EcoRI; line 4: DNA of phage B3 with EcoRI; line 5: DNA of phage A7 with HindIII; line 6: DNA of phage A8 with HindIII; and line 7: DNA of phage B3 with HindIII. (**b**) Profile of digestion with HinfIII. Line 1: standard of molecular weight 1 kpb; line 2: DNA of phage A7; line 3: DNA of phage A8; line 4: DNA of phage B3; and line 5: standard of molecular weight. (**c**) Digestion of genetic material with nucleases S1. Line 1: genetic material of phage A4; line 2: genetic material of phage A4; line 3: genetic material of phage A5; line 4: genetic material of phage A5; line 5: genetic material of phage A7; line 6: genetic material of phage A8; line 7: genetic material of phage B3; line 8: empty; and line 9: standard of molecular weight 1 kpb.

**Figure 3 foods-11-01164-f003:**
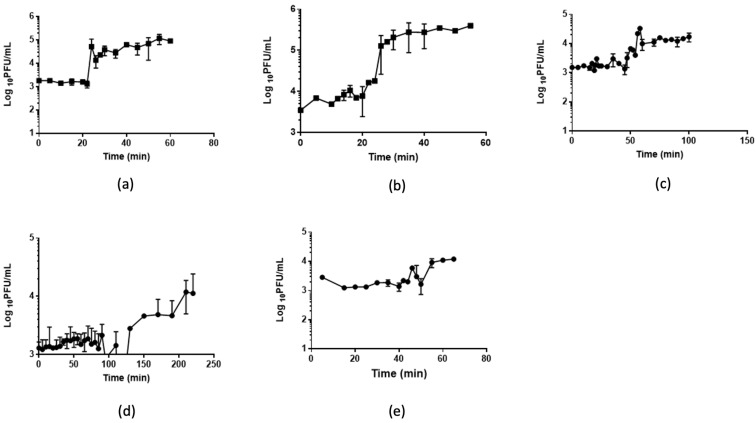
Growth kinetics. The figure shows the single-step growth curves of (**a**) phage A4, (**b**) phage A5, (**c**) phage A7, (**d**) phage A8, and (**e**) phage B3. Data shown are the mean of three replicates ± SD.

**Figure 4 foods-11-01164-f004:**
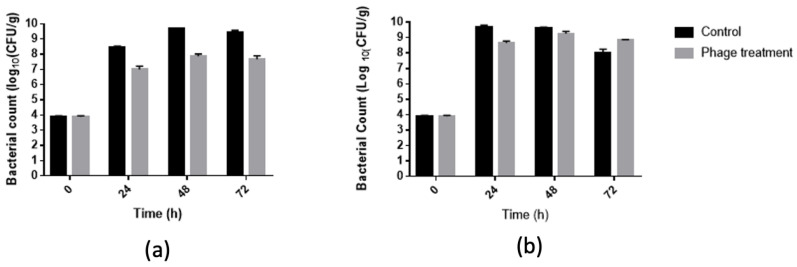
Evaluation of the antibacterial activity of phages cocktail in chicken meat to different temperatures (**a**) 22 °C and (**b**) 30 °C.

**Figure 5 foods-11-01164-f005:**
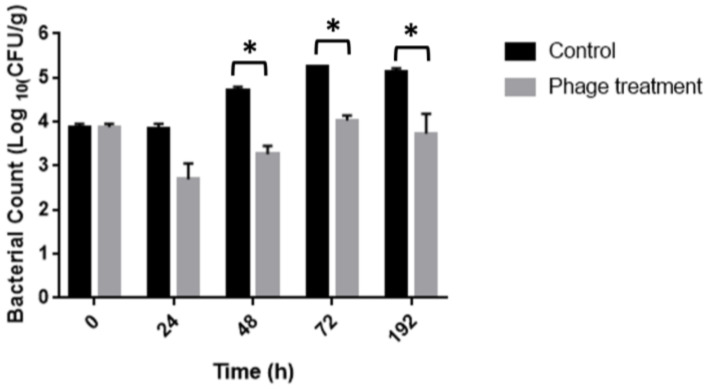
Evaluation of the antibacterial activity of phages cocktail in chicken meat to 10 °C. * *p* < 0.0001.

**Table 1 foods-11-01164-t001:** Host range of phage.

*Salmonella enterica* Serovar	Phage																				Total Phage
	A1	A2	A3	A4 *	A5 *	A5.2 *	A6	A7 *	A8 *	A9 *	A10	B1	B2	B3 *	B4	B5	B6 *	B7	B8 *	B9	
Choleraesiu FSL R9-1343 ^a^	3	3	3							3	3	3	3		3			3		3	10
Javiana FSL S5-406 ^a^											3	3									2
Infantis FSL S5-506 ^a^	3	3	3								3		3		3	3		3	3	3	10
Typhimuriu FSL A4-737 ^a^	3	3									3	3	3		3	3		3		3	9
Agora FSL S5-667 ^a^	3	3	3								3	3			3	3				3	7
Cerro FSL R8-370 ^a^							3	3	3	3		3		3			3	3	3		9
Newport FSL S5-515 ^a^		3	3																		2
4,5,12, i:- FSL S5-390 ^a^	3	3	3								3	3	3		3			3		3	8
Enteritidis SARB16 ^b^							3				3										2
Enteritidis SARB17 ^b^							3	3			3	3									4
Enteritidis SARB19 ^b^		3					3				3										4
Typhimuriu ATCC 14028	3	3	3	3	3	3	3	3	3	3	3	3	3	3	3	3	3	3	3	3	20
Total serovars	6	8	6	1	1	1	5	3	2	3	10	8	5	2	6	4	2	6	3	5	

*Salmonella* strains susceptible to phage infection are indicated by an orange box. ^a^ ILSI North America Collection; ^b^
*Salmonella* reference collection B SARB. *** Phages were classified as “narrow” host-range.

**Table 2 foods-11-01164-t002:** Kinetics parameters of phage infection in *Salmonella* Typhimurium ATCC 14028.

Phage	Burst Size (log 10 Unit)	Lag Time (min)
A4	4.9	20
A5.1	2.5	20
A7	1.8	>60
A8	1.7	>100
B3	0.8	4

## Data Availability

Data is contained within the article.

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
