# Peer review of "Use of Cocktail of Bacteriophage for Salmonella Typhimurium Control in Chicken Meat"

_foods, 2022, doi:10.3390/foods11081164_

Round 1
Reviewer 1 Report
Referee Report
The manuscript submitted by Aguilera and co-authors entitled “Use of cocktail of bacteriophage for Salmonella Typhimurium control in chicken meat” describes the significance of lytic bacteriophages cocktail in reducing the Salmonella Typhimurium colonies in the chicken meat. The manuscript requires the whole genome sequencing data to be incorporated in the paper before its approval for publication (A4, A5.7, A7, A8 and B3). It is very important to be sure about the biological nature of the virus (as describes the ICTV) system and then move for the application of bacteriophage cocktail and related downstream assays.
In introduction and methods section there is a full stop and then reference number is mentioned while in discussion section reference number and then full top is followed. Kindly follow the guidelines provided by the journal and adhere to those guidelines.
English writing needs drastic improvements.
Moreover, I have highlighted the sentences with blue highlights were restructuring of the sentences is required in abstract and introduction section.
In materials and methods section:
1.Please provide a statement in methods section to confirm that all methods were carried out in accordance with relevant guidelines and regulations.
2. Please provide a statement in methods section to confirm that all experimental protocols were approved by a named institutional and/or licensing committee/s.
- Line #70 LB stands for? Luria Broth (LB) or Lysogeny Broth.
- line #82. Kindly add space
- line #84. SM buffer stands for what?
- line #89 use μl instead of UL
- Figure 2 part c: picture quality is not good it should be replaced with a better figure.

Author Response
Dear reviewer:
We thank the commentaries and suggestions, we are sure that it will improve the quality of our work.
Please find below the answer to the comments.
Q1.1: The manuscript requires the whole genome sequencing data to be incorporated in the paper before its approval for publication (A4, A5.7, A7, A8 and B3). It is very important to be sure about the biological nature of the virus (as describes the ICTV) system and then move for the application of bacteriophage cocktail and related downstream assays.
R1.1: We partially agree with the opinion of the reviewer. The genome sequence of the phages isolated in this work undoubtedly will allow the classification of the viruses, and currently is ongoing in our lab. So, according to ICTV the identification and classification of viruses require a combination of approaches that includes the molecular composition of the genome; the structure of the virus capsid and whether or not it is enveloped; the gene expression program used to produce virus proteins; host range; pathogenicity; and sequence similarity. Thus, although the genome characterization if widely used now, this is only a part of the scheme of classification. Our approach includes composition of the genome, the structure of the virus capsid and whether or not it is enveloped (by electron microscopy), and host range, taking in account 3/5 of the criteria used for viral classification. By other hand, the goal of the articles is to prove the concept that the combination of different phage could be a useful tool to reduce the bacterial load of Salmonella. The future application of this cocktail will require the specific identification of the phage present in the cocktail and also the specific proportion of them, and adjustment of each food matrix considering their production line. These furthers characterization are currently by performed in our lab however we consider that are beyond of the scope of the food journal.
Q1.2: In introduction and methods section there is a full stop and then reference number is mentioned while in discussion section reference number and then full top is followed. Kindly follow the guidelines provided by the journal and adhere to those guidelines.
R1.2: The discussion and introduction was changes according to the journal guidelines.
Q1.3: English writing needs drastic improvements.
R1.3: We improved the English writing with an expert.
Q1.4: Moreover, I have highlighted the sentences with blue highlights were restructuring of the sentences is required in abstract and introduction section.
R1.4: We improved the English writing with an expert and take the suggestion in the sentences with blue highlights.
Q1.5: Please provide a statement in methods section to confirm that all methods were carried out in accordance with relevant guidelines and regulations.
R1.5: We incorporated a statement in methods section to confirm that all methods were carried out in accordance with SEREMI and the biosecurity committee of the University of Santiago of Chile. Lines: 72-75. “All experiments involving the manipulation of Salmonella enterica serotype Typhimurium ATCC14028 were performed in the CECTA laboratory under the ethical and biosafety standards of the Institutional Ethics Committee of the Universidad de Santiago de Chile (335) and current legislation on the subject (1613328729).”
Q1.6: Please provide a statement in methods section to confirm that all experimental protocols were approved by a named institutional and/or licensing committee/s.
R1.6: We incorporated a statement in methods section to confirm that all methods were carried out in accordance with SEREMI and the biosecurity committee of the University of Santiago of Chile. Lines: 72-75.” All experiments involving the manipulation of Salmonella enterica serotype Typhimurium ATCC14028 were performed in the CECTA laboratory under the ethical and biosafety standards of the Institutional Ethics Committee of the Universidad de Santiago de Chile (335) and current legislation on the subject (1613328729).”
Q1.7: Line #70 LB stands for? Luria Broth (LB) or Lysogeny Broth.
R1.7: The line#70 was changed from “ The strains were stored at -80°C in Luria medium supplemented with 50% w/v of glicerol… “ to “The strains were stored at -80°C in Luria-Bertani broth (LB) supplemented with 50% w/v of glicerol…” in line 68
Q1.8: line #82. Kindly add space, line #84. SM buffer stands for what?, line #89 use μl instead of UL, Figure 2 part c: picture quality is not good it should be replaced with a better figure.
R1.8:The line#82 was changes from “according to the double agar overlay method. 17,18Plates were incubated …“ to “according to the double agar overlay method [17,18]. Plates were incubated ” in line 87.
The line#84 was changes from “using a truncated pipette tip, suspended in 200 μl SM buffer “ to “and resuspending the plaque in 200 μL SM buffer (Tris-HCl 50 mM, NaCl 100 mM, MgSO4 x 7 H2O 8 mM pH7.5)” in line 90
The line#89 was changed from “The host range of the phages was tested by spotting 5 uL of the lysate (105 PFU/mL), “ to “The phage host-range was tested by spotting 5 µL of the lysate (105 PFU/mL), in” in line 94.
Figure 2 part c was changed for a picture of a better quality.
On behalf of the authors, VG
Regards

Reviewer 2 Report
figure 1, especially the 1a part should be improved, this is low quality. maybe you will find a better photo.
an interesting finding is the highest effectiveness of phages in 22*C You should more elaborate this in the discussion, I found this paper maybe could help: https://www.mdpi.com/1422-0067/22/21/11381 ; https://onlinelibrary.wiley.com/doi/pdf/10.1111/jfs.12747
at the end of the discussion, there is a lack of usage perspectives of these phages, they could be used for salmonella control in other foods (https://journals.plos.org/plosone/article?id=10.1371/journal.pone.0262946), environmental control (https://www.sciencedirect.com/science/article/abs/pii/S0929139316302785), treatment of infected animal or salmonella count reduction (https://www.ncbi.nlm.nih.gov/pmc/articles/PMC6211228/)
Author Response
Dear reviewer:
We thank the commentaries and suggestions, we are sure that it will improve the quality of our work.
Please find below the answer to the comments.
Q2.1: Figure 1, especially the 1a part should be improved, this is low quality. maybe you will find a better photo.
R2.1: Figure 1 part a was changed for a picture of better quality picture.
Q2.2:an interesting finding is the highest effectiveness of phages in 22°C You should more elaborate this in the discussion, I found this paper maybe could help: https://www.mdpi.com/1422-0067/22/21/11381 ; https://onlinelibrary.wiley.com/doi/pdf/10.1111/jfs.12747
R2.2: The discussion was modified and the comments of the reviewer incorporated. Lines 574-672.
Q 2.3at the end of the discussion, there is a lack of usage perspectives of these phages, they could be used for salmonella control in other foods (https://journals.plos.org/plosone/article?id=10.1371/journal.pone.0262946), environmental control (https://www.sciencedirect.com/science/article/abs/pii/S0929139316302785), treatment of infected animal or salmonella count reduction (https://www.ncbi.nlm.nih.gov/pmc/articles/PMC6211228/)
R2.3: The discussion was modified and the comments of the reviewer incorporated. Lines 574-672.
On behalf of the authors, VG
Regards

Round 2
Reviewer 1 Report
There are no further advised changes at my end in current draft
Reviewer 2 Report
authors properly correct all my suggestions